# Infiltration-Friendly Agroforestry Land Uses on Volcanic Slopes in the Rejoso Watershed, East Java, Indonesia

**Didik Suprayogo [1,*], Meine van Noordwijk [1,2,3]** **, Kurniatun Hairiah [1], Nabilla Meilasari [1], Abdul Lathif Rabbani [1], Rizki Maulana Ishaq [1] and Widianto Widianto [1]**

[1]   Soil Science Department, Faculty of Agriculture, Brawijaya University, Jl. Veteran no 1, Malang 65145, Indonesia; m.vannoordwijk@cgiar.org (M.v.N.); K.HAIRIAH@CGIAR.ORG (K.H.); nabilla.meilasari@gmail.com (N.M.); agroforestry@ub.ac.id (A.L.R.); rizky.maulana17@gmail.com (R.M.I.); widianto@gmail.com (W.W.)

[2]   World Agroforestry Centre, ICRAF, Indonesia Office, Bogor 16001, Indonesia

[3]   Plant Production Systems, Wageningen University, 6708 PB Wageningen, The Netherland

\*   Correspondence: Suprayogo@ub.ac.id or Suprayogo09@yahoo.com

**Abstract:** Forest conversion to agriculture can induce the loss of hydrologic functions linked to infiltration. Infiltration-friendly agroforestry land uses minimize this loss. Our assessment of forest-derived land uses in the Rejoso Watershed on the slopes of the Bromo volcano in East Java (Indonesia) focused on two zones, upstream (above 800 m a.s.l.; Andisols) and midstream (400–800 m a.s.l.; Inceptisols) of the Rejoso River, feeding aquifers that support lowland rice areas and drinking water supply to nearby cities. We quantified throughfall, infiltration, and erosion in three replications per land use category, with 6–13% of rainfall with intensities of 51–100 mm day$^{-1}$. Throughfall varied from 65 to 100%, with a zone-dependent intercept but common 3% increase in canopy retention per 10% increase in canopy cover. In the upstream watershed, a tree canopy cover > 55% was associated with the infiltration rates needed, as soil erosion per unit overland flow was high. Midstream, only a tree canopy cover of > 80% qualified as "infiltration-friendly" land use, due to higher rainfall in this zone, but erosion rates were relatively low for a tree canopy cover in the range of 20–80%. The tree canopy characteristics required for infiltration-friendly land use clearly vary over short distances with soil type and rainfall intensity.

**Keywords:** entrainment; erosion; forest conversion; overland flow; soil macroporosity; throughfall; water balance

## 1. Introduction

Water access for all, the Sustainable Development Goal 6 of the Agenda 2030 agreed by the United Nations [1], not only refers to drinking water and sanitation. It requires the protection of "infiltration-friendly" land uses in upland watersheds as a source of clean water [2]. Sufficient groundwater recharge is important to the sustainable management of groundwater resources to maintain streamflow throughout the year, as well as to feed springs [3,4]. While much of the public discourse is in terms of forest versus agriculture, thresholds for specific soil and climate regimes are needed within the intermediate agroforestry spectrum of land uses [5]. Thresholds to critical hydrological functions are likely dependent on local context but need to be understood to guide natural resource management in the challenging trade-offs between local and external priorities [6]. Hydrological functions, and their sensitivity to climate change, can be characterized by a number of metrics [7]. Much of the literature on forest hydrology is concerned with reductions in annual water

yield due to increased canopy interception and/or tree water use by fast-growing forest stands [8], without a distinction between (fast) overland and (slower, infiltration-dependent) subsurface flow pathways. A recent review [9] found that the recovery of annual river flow with the age of planted forest is an exception rather than a rule. However, the recovery of infiltration with tree cover can increase dry season flows [10] without increasing annual water yield. Changes in streamflow regime will reflect both changes in evapotranspiration (ET) and in infiltration after the change in land use under given climatic conditions [11,12]. On the other hand, a high tree (*Acacia auriculiformis*) canopy, without an understory and permanent litter layer, was associated with high erosion rates due to high-impact drips from the leaves [13]. Empirical data and process-level understanding is needed of these diverse and partly contradictory effects of tree cover, especially in human-managed land uses.

Agroforestry systems with high canopy densities can, if a permanent litter layer is present, maintain high infiltration rates and can positively impact on hydrologic functions through: (1) a green canopy cover at the tree and understory level, (2) land surface roughness, (3) litter at the soil surface, and (4) water uptake by trees and other vegetation [14,15]. Five aspects that hydrologically differentiate natural forest from open-field agriculture, with intermediate functionality for managed forest, plantations, agroforestry, and trees outside forest [16,17], are: (1) the leaf area index (LAI) that allows photosynthesis when stomata are open and transpiring, and that, along with leaf morphology and rainfall intensity, determines canopy interception, retention, and subsequent evaporation, (2) the surface litter that prevents crusting and supports infiltration [18] while reducing soil evaporation and reduces the entrainment of soil particles if overland flow still occurs, (3) the soil macroporosity that governs infiltration and allows for the aeration of deeper soil layers between rainfall events while recovering at a decadal time scale after reforestation [19,20], (4) the root systems that govern water extraction from deeper soil layers, in conjunction with the phenology of the aboveground canopy [21], and (5) possible influences on rainfall events [22,23]. Each of these five aspects has its own dynamic (time constants) and dependency on the type of trees and their management, challenging the definition of hydrologically adequate land use choices. Rather than prescribing, independent of soil types and slope, the type and quantity of tree cover that is needed, as tends to happen in forest zoning, it may help if limits to infiltration-friendly land use (focused on the third function) can be operationalized in a local context. In terms of watershed hydrology, infiltration-friendly land uses can be interpreted as any land use that allows high rates of water infiltration so that surface runoff is a small (to be defined in local context) fraction of rainfall and the watershed functions of flow buffering and erosion control are secured (to specified standards). River flow in watersheds that provide perfect buffering might theoretically be constant every day, but in practice, a "flow persistence" metric of about 0.85 is hard to surpass [24]. Flow buffering is essential for climate resilience [25] and high flow persistence metrics are desirable, as they directly relate to peak flow transmission [26].

The discussion on forests and watershed functions in Indonesia became based on specific theories about underlying mechanisms and measurements in the 1930s [27–30], but at the policy level, a generic dichotomy between "forest" and "non-forest" conditions was maintained. The Indonesian spatial planning law prescribes that 30% forest cover is needed in all local government entities to secure hydrological forest functions [31]. As the 30% norm originated in studies of flooding risk in relation to spring snow melt in Switzerland [32], a more nuanced and process-based understanding is needed to underpin effective policies on desirable forest cover, especially in densely populated Java, where agroforests are common. In Southeast Asia, 8.5% of the global human population lives on 3.0% of the land area. With 7.9% of the global agricultural land base, the region has 14.7% and 28.9% of such land with at least 10% and 30% tree cover, respectively, and is the world's primary home of "agroforests" [33].

On densely populated Java, volcanic slopes are home to large numbers of farmers, while also serving as sources of water for lowland agriculture and the rapidly growing cities. The shrinking area of state-managed forests is no longer able to secure the required watershed functions, but at least part of the agroforestry managed by farmers can meet the required hydrological functions [34].

In the Rejoso Watershed (Pasuruan, East Java, Indonesia), numerous stakeholders depend on the watershed functions of densely populated mountain slopes to meet their water demand. These include local communities, farmers using water irrigation, the Regional Water Company, and bottled water industries. A major infrastructure is planned to bring water to Surabaya and the surrounding urban centers. However, the quantity and quality of the water at the source of the pipe have been decreasing over the past 10 years, putting the infrastructure investment at risk. Decreasing water resources are likely due to land use changes in the recharging area of the Rejoso Watershed on the northern slopes of the Bromo-Semeru volcanic mountain range, and/or decreased pressure on artesian wells across the land due to increased extraction for paddy rice fields. Among the hydrologic functions, infiltration is critical, as water travels through the subsoil to artesian wells at the foot of the volcano, in addition to surface rivers.

This research in the Rejoso Watershed using runoff and soil erosion plot-scale studies under natural rainfall [35] within the locally relevant range of land cover types was thus designed to assess which land uses can maintain infiltration rates under local peak rainfall intensity and restrict soil erosion to acceptable levels. The specific questions were:

1. Which existing land uses limit infiltration below the required rates at peak rainfall events?
2. Which factors that are directly observable, such as tree cover, litter layer thickness, or surface roughness, can be used to define thresholds for "infiltration-friendly land use"?
3. Do the answers to questions 1 and 2 need to be differentiated between the upper and middle watershed, with current vegetable production and agroforestry as dominant land uses, respectively?

## 2. Materials and Methods

### 2.1. Study Area

The Rejoso Watershed, is located on the northern slope of Mount Bromo, covering 16 sub-districts in Pasuruan District, East Java Province, Indonesia. The Rejoso Watershed is located between 7°37′13.35″ and 7°55′18.63″ South, and between 112°48′32.51″ to 113°55′55″ East (Figure 1).

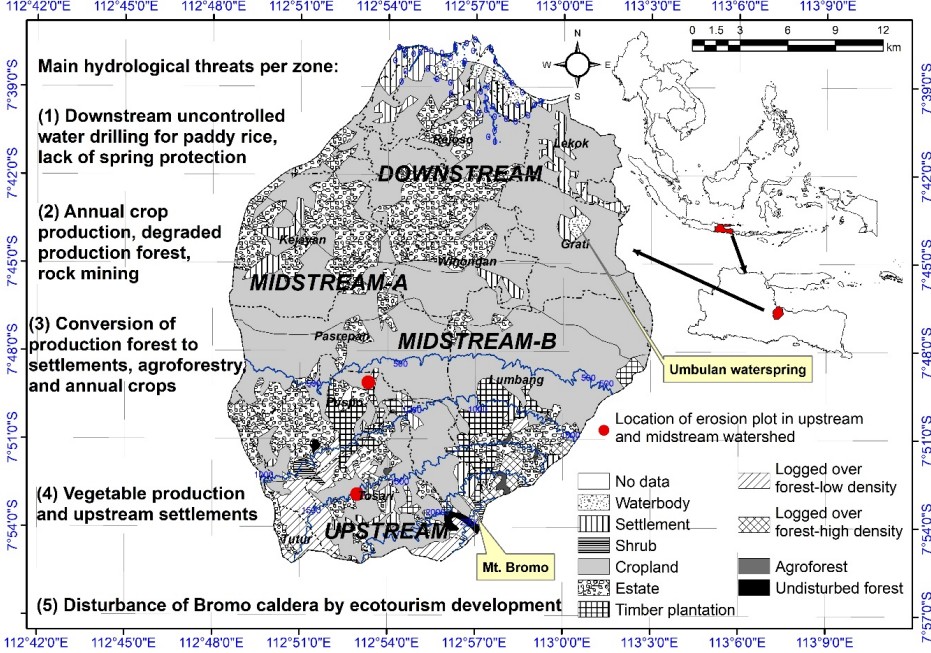

**Figure 1.** A. Position of the Rejoso Watershed in East Java as part of the Indonesian archipelago in Southeast Asia. B. The Rejoso Watershed from upstream (at the bottom) to sea level and land uses considered to be a hydrological threat; purple indicates open soil, green tree cover (base map in [36]).

The Rejoso Watershed covers an area of 634 km$^2$ with a hydrologic (watershed) length of the main channel of about 22 km. This study was conducted in two locations, namely in the upstream (above 800 m a.s.l.) and midstream (400–800 m a.s.l.) sections, with the dominant vegetation (land cover) selected for each location (Figure 1).

Climatic conditions that influence hydrology and erosion are largely determined by the influence of the northwest and southwest monsoons. The northwest monsoon, picking up large amounts of moisture over the Indian Ocean, brings in most of the annual precipitation in the area, and predominates during the period from November through April. Although there is considerable variation in the amount and distribution of rainfall from year to year, most places in the watershed receive about 91% of the rainfall during the November–May wet season (monthly rainfall > 100 mm) in the upper stream and about 91% of the rainfall during the November–April wet season in the midstream (Figure 2). Due to topographic influences, there is considerable spatial variation in annual precipitation as well, ranging from 1655 mm to 3675 mm, with extreme yearly rainfall in 2010, with an annual precipitation of 5298 mm over 24 years of rainfall measurements (1990–2013) in the upper stream compared with an annual precipitation ranging from 1020 mm to 2603 mm in the midstream. The May to October period is considered the dry season. Then the southeast monsoon predominates, bringing much smaller amounts of precipitation due to the lower atmospheric moisture caused by lower temperatures in the Southern Hemisphere at this time of the year. The annual precipitation in the upstream (average annual precipitation = 2488 mm) is higher than the in midstream (average annual precipitation = 1632 mm). Over 24 years of measurements, the maximum daily rainfall in the upper stream and midstream ranged from 80 mm day$^{-1}$ to 200 mm day$^{-1}$ and 60 mm day$^{-1}$ to 320 mm day$^{-1}$, respectively. Based on Schmidt–Ferguson climate classification, the upper stream and midstream are considered rather wet (C) and average (D), respectively.

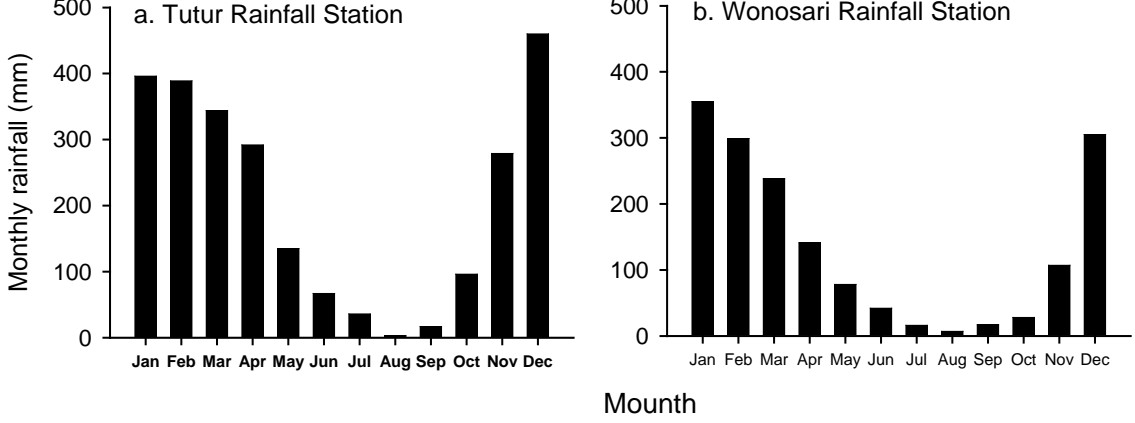

**Figure 2.** Monthly rainfall distribution in the Rejoso Watershed from the average of 24 years of rain events (1990–2013) in the (**a**) upstream (Tutur Rainfall Station) and (**b**) midstream (Wonosari Rainfall Station).

The Rejoso Watershed consists of four types of soil, namely: Andisols, Inceptisols, Alfisols, and Entisols. Andisols are mainly found on the upper slopes of the volcano. Andisols have a distinct black to very dark brown surface horizon rich in organic matter, which usually overlies a brown to dark yellowish-brown subsoil. The clay fraction is dominated by allophane. Andisols are highly permeable, porous with low bulk density, have high water-holding capacity, and a crumb structure. The most common texture is sandy loam. These soils have high inherent fertility and are highly erodible only when seriously disturbed. The middle and some lower volcanic slopes, consisting of easily weatherable permeable tuffs and ash deposits, give rise to deep soils—Inceptisols and Alfisols. Inceptisols have only limited horizon differentiation. Their texture ranges from deep friable clays to clay loams. Alfisols are soils which have an accumulation of clay in the subsoil. Their texture ranges from loam to clay loam in the topsoil and clay loam to clay in the subsoil. Both soils have moderate to high inherent

fertility but are highly susceptible to erosion. The fourth group, Entisols, are soils that lack horizon development and are found on volcanic sands, ashes, and tuffs. Entisols occur on recent and sub-recent lahars of the Bromo volcano. Entisols with a coarse texture are extremely erodible and have very low water-holding capacities. Permanent vegetative cover, and especially diversified tree crops and agroforestry or forestry, are the most suitable land utilization types to prevent erosion.

## 2.2. Land Cover Types Compared

In important research traditions associated with the Universal Soil Loss Equation (USLE) [37], the quantification of erosion requires a "bare soil" reference, expressing the degree of protection provided by vegetation relative to this "control". Fortunately, bare soil is rare in this landscape, and it would be considered an extreme, rather than a standard agricultural point of reference. Artificially clearing land to allow such treatment to be measured would give results that are hard to be interpreted, as soil changes after clearing would lead to a time-dependence of the results, rather than being an unambiguous point of reference. By referring to the more process-based Rose equation [38], separating overland flow as a transport medium and "entrainment" as a soil characteristic relative to the energy-dependent transport capacity of such flow, we do not depend on the USLE framework (that infers that soil loss is universal, but does not account for its counterpart process, sedimentation [39]) but can focus on existing land covers and associated land uses in the landscape.

In both the upstream and midstream parts of the catchment, four dominant land use systems were assessed (Table 1), spatially replicated in three separate measurement plots. Upstream land uses included old and young pine plantations (production forest) and highland vegetable crops with variations in tree canopy cover in the landscape on steep (30–60%) to very steep (>60%) land with imperfect ridge terraces. Midstream land uses included production forest, multistrata coffee-based agroforestry, clove-based agroforestry, and several mixed agroforestry types with variations in tree canopy cover in the landscape on moderately steep (15–30%) and steep (30–60%) land with bench terraces sloping outward.

**Table 1.** Land use, vegetation, soil conservation measure, and slope of measurement plots.

| Code | Land Use | Vegetation (the Average Height of Trees) | Terracing | Slope (Plot Level, %) |
|---|---|---|---|---|
| | | **Upstream Rejoso Watershed** | | |
| UT1 | Old production forest | Pine (*Pinus merkusii*) (34 m) + grass | None | 35–40 |
| UT2 | Young production forest | Pine (11 m) + grass | None | 50–60 |
| UT3 | Agroforestry | Strip cemara (*Casuarina junghuniana*) (13 m) + cabbage | None | 40–50 |
| UT4 | Arable land | Banana, maize, carrot | None | 40–50 |
| | | **Midstream Rejoso Watershed** | | |
| MT1 | Old production forest | Mixed pine (28 m) or mahogany (*Swietenia macrophylla*) (12 m), banana, salak (*Salacca zalacca*), taro (*Colocasia esculenta*), elephant grass (*Miscanthus giganteus*). | Bench terrace sloping outward | 3–8 |
| MT2 | Agroforestry | Coffee-based (2 m) mix with durian (*Durio zibethinus*) (10 m), mahogany (9 m), *Leucaena leucocephala* (8 m), *Paraserianthes falcataria* (11 m), *Albizia saman* (11 m), dadap (*Erythrina variegata*) (11 m), banana | Bench terrace sloping outward | 3–8 |
| MT3 | Agroforestry | Clove (*Syzygium aromaticum*) (8 m), banana | Bench terrace sloping outward | 3–8 |
| MT4 | Agroforestry | Mango (*Mangifera indica*) (10 m), durian (10 m), *Randu kapuk* (*Ceiba pentandra*) (11 m), maize, cassava, groundnut | Bench terrace sloping outward | 3–8 |

*2.3. Quantifying Terms of Water and Soil Balance*

2.3.1. Overview

As forest and tree cover can influence various steps in the chain from rainfall to streamflow and erosion, we aimed to quantify (1) the direct effect of tree canopies on the retention of part of the rainfall (followed by direct evaporation), versus the fraction reaching the soil surface by throughfall and stemflow, (2) the partitioning of the latter into infiltration and overland flow, (3) the entrainment of soil particles into this overland flow. Within the time and resources available we did not assess (4) the seepage of groundwater beyond the root zone and access by vegetation, (5) the pathways and release of groundwater into streams, (6) the routing of overland flow into streams, or (7) the in-field (or riparian filter zone) sedimentation of entrained soil particles, beyond the scale of the measurement plots. However, we did characterize the vegetation and soil characteristics that influence the various processes.

2.3.2. Rainfall and Throughfall

Rain gauges, outside of the direct influence of tree canopies, were installed in four observation locations (with adjacent erosion plots) upstream and four observation locations midstream of the Rejoso Watershed. In each runoff plot, throughfall was measured with five replications. The throughfall gauges below the tree canopy had a horizontally placed 30 cm diameter funnel 120 cm above the soil surface and a collector bottle with a volume of 1.5 dm$^3$ placed with bamboo as a support. Throughfall and rainfall were collected every day for two months of the rainy season, from March to May 2017. Attempts were made to also quantify stemflow, but due to technical problems with the method used, no reliable data were obtained, and the results are not shown here.

2.3.3. Water Infiltration and Soil Erosion Measurement

Water infiltration was quantified in each land cover type via its complement, surface runoff, and expressed in the runoff/throughfall ratio. As the throughfall for quantified infiltration was measured below the tree canopy, the amounts were the net of canopy retention and possible stemflow. Surface runoff was measured in 6 m × 2 m plots protected from surface run-on, with two drums at the lower end to collect surface runoff and sediment concentrations for soil erosion measurements (Figure 3).

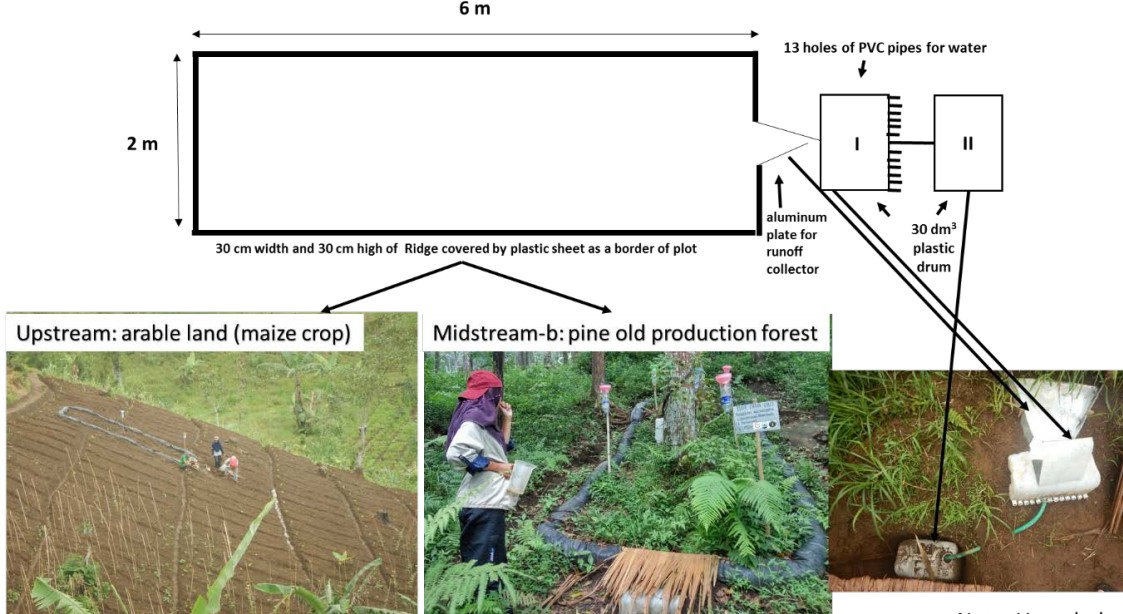

**Figure 3.** Runoff and soil erosion plot design.

In each plot, the water flow was collected into two collection drums with a capacity of 30 dm$^3$. The first drum had a divider system channeling into 13 channels (PVC pipes) with equal diameters and level positions, with one connected with a second drum. The volume of water flowing from each pipe was measured to calibrate the water volume proportion entering into the second drum. The potential capacity of the runoff collector thus was (30 dm$^3$ * 13) + 30 dm$^3$ = 420 dm$^3$ for 12 m$^2$ or 35 mm. We did nor encounter situations where the second tank overflowed. Runoff samples at each plot were collected every day and the rain that occurred during the measurement period was measured by measuring the water depth in each drum. The amount of runoff in each rain event was calculated using Equations (1) and (2):

$$R_t = V_{d-I} + (13 * V_{d-II})$$ (1)

$$V_d = 1000 * (D * L * W)$$ (2)

where $R_t$ is total runoff (dm$^3$), $V_d$ is the water volume in drums I and II (dm$^3$), L = length and W = width of drum (cm), and D is the water depth in each drum (cm). The total runoff was then divided by the area of the plot (2 m × 6 m) to convert to mm. Data could be compared to a classification developed elsewhere [40] that indicates a runoff coefficient of 0.14 as adequate for Andisols, and 0.20 for Inceptisols.

Soil erosion in each rain event was determined by collecting 1 dm$^3$ of runoff sediment in each drum. The sample was filtered with "newsprint" and dried in an oven with a temperature of 105 °C to get the weight of the sediment (S). In earlier studies, we found that effluent from this readily available filter material had a negligible sediment concentration [41]. Erosion (E) in each rain event was calculated using Equation (3):

$$E = ((V_{d-1} * S) + (13 * (V_{d-2} * S))) * \left(\frac{10^{-2}}{A}\right)$$ (3)

where E is soil erosion (Mg ha$^{-1}$), S is sediment (g dm$^{-3}$), and A is the area of the plot (m$^2$).

*2.4. Determination of Soil Properties*

Three bulk mineral soil samples were collected from each layer of soil of 0–10 cm, 10–20 cm, 20–30 cm, 30–40, and 40–50 cm for soil texture analysis and each layer of soil of 0–10 cm, 10–20 cm, and 20–30 cm for soil bulk density, particle density, total soil porosity, and soil organic matter content. Particle size distribution (particles < 2 mm) was determined with the Bouyoucos densimeter method [42] after $H_2O_2$ pre-treatment and after samples had been dispersed in 5% sodium hexametaphosphate and 5% dispersing solution. Bulk density (oven dry weight per unit volume) was measured for a block-sized sample (20 cm × 20 cm × 10 cm = 4000 cm$^3$) collected in field moisture conditions (modified from [43]). Particle density was measured by the pycnometer method. Total soil porosity (∅), the percentage of the total soil volume that is not filled by solid (soil) particles [44], was calculated from bulk density data and particle density using Equation (4):

$$\varnothing = \left(1 - \frac{\rho_b}{\rho_p}\right) \times 100\%$$ (4)

where ∅ is porosity (%), $\rho_b$ is bulk density (g cm$^{-3}$), and $\rho_p$ is particle density (g cm$^{-3}$).

Soil organic carbon (SOC) was determined by dichromate oxidation [45]. Soil infiltration was measured by the standard double-ring infiltrometer test [46]. The double-ring infiltrometer as often constructed from a thin-walled steel pipe with inner and outer cylinder diameters of 20 and 30 cm, respectively.

The soil macro-porosity was measured using the methylene blue method, by looking at the blue distribution pattern of the methylene blue solution in the soil profile. The methylene blue solution (70 g methylene blue per 200 L of water) was gradually poured into the ground, which had been bound

by a metal frame measuring 100 cm × 50 cm × 30 cm (Figure 3) and left for 36 h until the methylene blue solution soaked into the soil. Methylene blue will pass through soil macropores but be absorbed by micropores and soil surfaces. After all the methylene blue solution had disappeared from the soil surface, the top 5 cm of soil was removed from a 100 cm × 100 cm sample area and infiltration patterns were recorded, before a further 10 cm of soil was removed for a second map (at 15 cm below the soil surface), and a further 10 cm of soil for a third map (25 cm below soil surface). For mapping blue patches in each horizontal plane, a transparent sheet of plastic was placed on the surface and all visible blue patches were mapped with marker pens. Blue distribution patterns, redrawn on tracing paper, were photocopied for analysis of the black-and-white pattern of the fraction of soil involved in macropore flow with the IDRISI computer program.

*2.5. Other Plot Characteristics*

2.5.1. Canopy Cover

The canopy cover can be defined as the percentage of tree canopy area occupied by the vertical projection of tree crowns [47]. The percentage of canopy cover is measured by scathing the shadow of sunshine at ground level using 10 m × 10 m sheets of white paper. The canopy projection when the sun was overhead was drawn to scale on white paper in each of the four quadrants of the 20 m × 20 m plots, after which the shaded areas were cut out and weighed separately. Canopy cover was calculated according to Equation (5):

$$\%\text{Canopy Cover} = \frac{\text{W Canopy}}{\text{W Total}} \times 100 \tag{5}$$

where %Canopy Cover is the percentage of tree canopy cover, W Canopy is the paper weight representing canopy cover and W Total is the paper weight representing the total area of observation, respectively.

2.5.2. Understory and Litter

Understory vegetation and litter were measured according to the rapid carbon stock appraisal protocol [48], using 50 cm × 50 cm samples for fresh weight, with subsamples dried for dry weight determination.

2.5.3. Land Surface Roughness

Surface roughness was measured in each plot as the standard deviation of elevation measured every 30 cm along a thread (thin rope) installed 30 cm from the surface vertically, horizontally, and diagonally over the erosion plot [49]. The measurement of the difference in elevation was set to a pixel size of 30 cm × 30 cm. Each plot was divided into six pixels for a 2 m plot width and 20 pixels for a 6 m plot length, so there were 120 pixels (N). Pixels were made on a flat plane 30 cm from the ground point of reference with a thin rope. In each center, the pixel was measured vertically parallel to the thin rope towards the surface of the ground with a ruler. The results of the measurements of height differences in each pixel were used to calculate Ra with the equation:

$$\text{Ra} = \frac{1}{N} \sum_{n=1}^{N} | h_n | \tag{6}$$

where N = Number of pixels in the patch and $h_n$ = difference of elevation between the nth pixel in the patch and the mean value.

*2.6. Data Analysis*

To answer the first research question, the null hypothesis was that within the forest to open field agriculture continuum of any observed difference in soil hydrological functions could be due to random variation. To see if that null hypothesis could be rejected, we examined differences in soil infiltration, runoff coefficient, and soil erosion between the dominant land uses in the upstream and midstream with Fisher's Least Significant Differences (LSD) test. Fisher's LSD test, which establishes differences between groups defined for independent samples, was used for hypothesis testing, given that the data met the requirements for normality and the homogeneity of variances. A probability level of 0.05 was set for rejecting the null hypothesis of no difference in tests of statistical significance. We used the GenStat 15th edition software for Fisher's LSD tests. The soil infiltration, runoff coefficient, and soil erosion were then compared with the soil infiltration category [50], existing infiltration adequacy standards, and acceptable soil erosion rates. An acceptability threshold, below which soil erosion is less than an "agriculturally permissible" rate ($E_{apr}$, Mg ha$^{-1}$ year$^{-1}$), was derived as:

$$E_{apr} = \left( \frac{\text{Depth of soil} * \text{Factor of soil depth}}{\text{Time horizon}} \right) * \text{Soil bulk density} \tag{7}$$

Both Andisols and Inceptisols are deep (beyond 120 cm soil depth) and have a soil depth factor of 1.0. We chose 400 years as a time horizon. Given the average soil bulk density of Andisols (0.83 g cm$^{-3}$) and Inceptisols (0.99 g cm$^{-3}$), we obtained $E_{apr}$ Andisol = 24.9 Mg ha$^{-1}$ year$^{-1}$ and $E_{apr}$ Inceptisol 29.7 Mg ha$^{-1}$ year$^{-1}$.

For the second research question, we tested a number of plot scale characteristics as possible indicators of "infiltration-friendly" plot characteristics: tree canopy cover, understory vegetation, litter necromass, and land surface roughness. Linear regression relationships between the surface runoff/rainfall ratio or soil erosion and the amount of rainfall, tree canopy cover, understory, litter, and land surface roughness were determined using SigmaPlot version 10.0. While a search for "explanatory" factors might have explored multiple regression, our focus was single indicators that could be used as proxies for follow-up discussions with farmers about adjusting land use.

The third research question required the analysis of data for the first two research questions, with the expectation that any thresholds for acceptable hydrological disturbance could be zone-specific, given variations in rainfall, soil type, and the specific characteristics of land use and vegetation.

## 3. Results

*3.1. Rainfall and Throughfall*

Within the measurement period, 31 rainy days were recorded (Figure 4). Rainfall variation between the upstream and midstream observation plots was relatively high, with an average of 520 mm (range 476–556 mm among 12 rain gauge measurements), and an average of 666 mm (range 541–840 mm among 12 rain gauge measurements), respectively. In the upstream and midstream areas, 71% and 57% of the rainy days had < 20 mm day$^{-1}$ ("light rain"), 24% and 31% had "moderate" rainfall (21–50 mm day$^{-1}$) and 6% and 13% "heavy" rain (51–100 mm day$^{-1}$), respectively; none had "very heavy rain" (>100 mm day$^{-1}$). Such rain conditions indicate that the rain erosivity in the midstream is higher than that of the upstream.

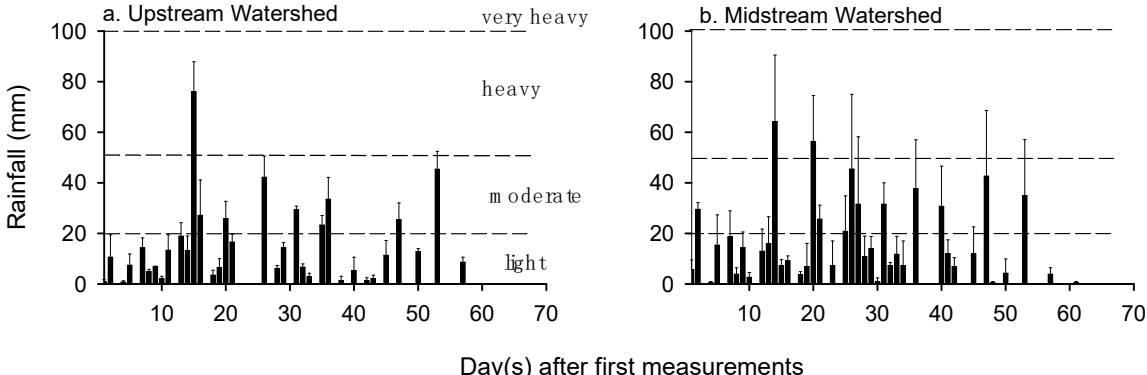

**Figure 4.** Distribution of rainfall during observation starting on March 03, 2017 in the Rejoso Watershed.

In the upstream area, the old production forest obtained a throughfall/rainfall ratio of 0.73 (standard deviation (SD) = 0.05), while for open-field agriculture it was 0.94 (SD = 0.5) (Figure 5a). For young production forests of *Casuarina junghuniana*-based agroforestry, the throughfall/rainfall ratio was 0.83 (SD = 0.05). In the midstream, the throughfall/rainfall ratio in agroforestry systems with tree canopies of 87%, 75%, and 52% were 0.81 (SD = 0.07) 0.81 (SD = 0.07), and 0.1 (SD = 06), respectively (Figure 5b). For agroforestry with low cover (26%), the throughfall/rainfall ratio was 0.96 (SD = 0.01).

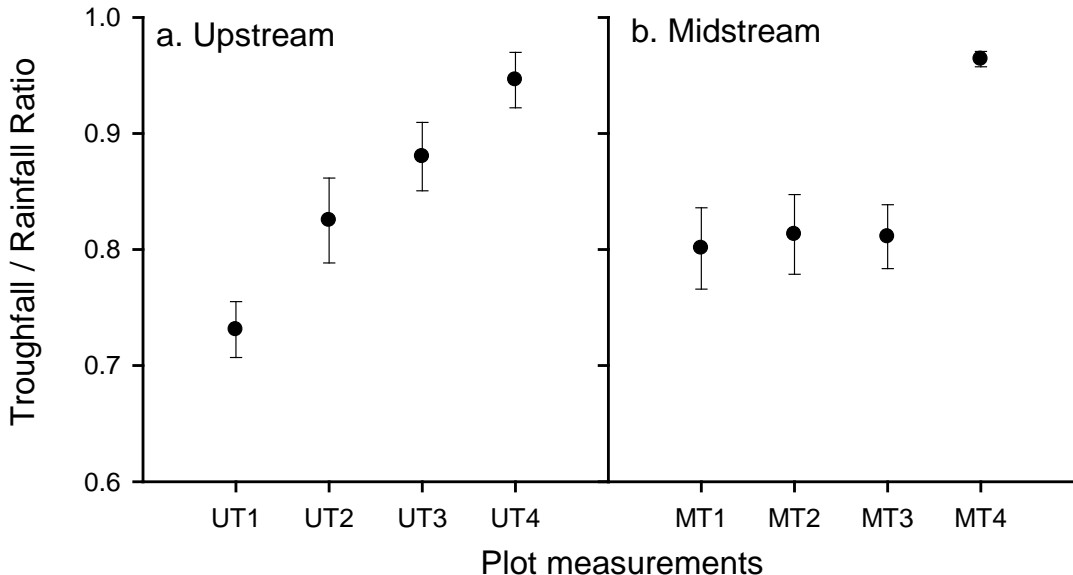

**Figure 5.** The throughfall/rainfall ratio variability in measured runoff plots (**a**) upstream, (**b**) midstream.

### 3.2. Soil Properties

The Andisols in the upstream area had a 40–60% silt fraction in all soil measured layers; the Inceptisols had a higher clay fraction (Appendix A). The upstream area had a lower bulk density and higher soil porosity, with a lower clay content than the midstream area (Table 2). The soil organic carbon content varied from 0.65 to 2.12%.

**Table 2.** Bulk density, particle density, soil porosity, macro-porosity, and organic C of runoff plots.

| Location Code at Soil Depth (cm) | Bulk Density (g cm$^{-3}$) * | | | Particle Density (g cm$^{-3}$) * | | | Soil Porosity (%) * | | | Soil Macro- Porosity (%) | | | $C_{org}$ (%) * | | |
|---|---|---|---|---|---|---|---|---|---|---|---|---|---|---|---|
| | 0–10 | 10–20 | 20–30 | 0–10 | 10–20 | 20–30 | 0–10 | 10–20 | 20–30 | 0–10 | 10–20 | 20–30 | 0–10 | 10–20 | 20–30 |
| Upstream Rejoso Watershed: Andisols | | | | | | | | | | | | | | | |
| UT1 | 0.87a | 0.81a | 0.83a | 2.16a | 2.23a | 2.31a | 60a | 63a | 64c | 8.0b | 5.2b | 0.9a | 2.05bc | 1.61c | 1.79b |
| UT2 | 0.85a | 0.86a | 0.82a | 2.27a | 2.30a | 2.33a | 63a | 63a | 65c | 5.1ab | 1.5a | 0.3a | 2.46c | 1.56bc | 1.78b |
| UT3 | 0.81a | 0.84a | 0.85a | 2.14a | 2.12a | 2.28a | 62a | 60a | 63b | 4.7ab | 2.1ab | 1.4a | 1.17a | 0.58a | 0.71a |
| UT4 | 0.84a | 0.88a | 0.84a | 2.28a | 2.29a | 2.08a | 63a | 62a | 60a | 3.0a | 0.3a | 0.1a | 1.35ab | 1.06ab | 0.92a |
| LSD | 0.07 | 0.13 | 0.12 | 0.17 | 0.21 | 0.38 | 4 | 5 | 1 | 3.52 | 3.4 | 1.8 | 0.85 | 0.50 | 0.50 |
| Midstream Rejoso Watershed: Inceptisols | | | | | | | | | | | | | | | |
| MT1 | 0.83a | 0.85a | 0.83a | 2.20a | 2.28a | 2.20a | 62c | 63a | 62b | 13.6ab | 7.0bc | 2.5c | 1.73a | 1.87a | 1.65b |
| MT2 | 0.96b | 0.91a | 0.91a | 2.42b | 2.38a | 2.21a | 60bc | 62a | 59ab | 16.1b | 8.3c | 1.8bc | 2.22a | 1.59a | 1.84b |
| MT3 | 1.03bc | 0.96a | 0.94ab | 2.38b | 2.36a | 2.40a | 57ab | 59a | 61b | 11.7a | 3.4ab | 0.9ab | 2.19a | 1.61a | 1.01a |
| MT4 | 1.09c | 1.04a | 1.04b | 2.38b | 2.33a | 2.33a | 54a | 55a | 55a | 11.4a | 0.8a | 0a | 1.71a | 1.36a | 1.12a |
| LSD | 0.10 | 0.24 | 0.11 | 0.15 | 0.17 | 0.22 | 4 | 10 | 4 | 4.0 | 3.9 | 1.0 | 0.84 | 0.54 | 0.41 |

* The same letter indicates no statistically significant differences between locations with Fisher's Least Significant Differences (LSD) test ($p < 0.05$).

### 3.3. Land Characteristics Related to Runoff and Soil Erosion

Production forests in the upstream area had a lower tree canopy cover than those midstream but higher than those in agroforestry systems (Table 3). Agroforestry in the upstream area had a very low tree canopy cover because trees were planted only along field edges. Midstream agroforestry gardens ranged from high (75%) to low (26%) canopy cover. Understory vegetation was more prominent upstream than midstream. Litter layer necromass and land surface roughness were generally aligned with tree canopy cover.

**Table 3.** Canopy cover, understory vegetation, litter necromass, and soil roughness of the sample plots.

| Code | Land Cover | Tree Canopy Cover (%) * | Understory Vegetation (Mg ha$^{-1}$) * | Litter (Mg ha$^{-1}$) * | Soil Roughness (%) * |
|---|---|---|---|---|---|
| *Upstream Rejoso Watershed* | | | | | |
| UT1 | Old production forest | 55b | 10.1b | 9.2b | 8.5a |
| UT2 | Young production forest | 40b | 10.5b | 2.0a | 7.0a |
| UT3 | Agroforestry | 4a | 10.1b | 2.1a | 9.5a |
| UT4 | Arable land | 0a | 3.7a | 0.3a | 7.7a |
| LSD | | 15 | 5.6 | 3.7 | 4.6 |
| *Midstream Rejoso Watershed* | | | | | |
| MT1 | Old Production Forest | 87c | 2.5a | 9.8b | 7.6b |
| MT2 | Agroforestry | 75c | 2.5a | 4.8a | 5.4ab |
| MT3 | Agroforestry | 52b | 2.1a | 5.2a | 2.8a |
| MT4 | Agroforestry | 26a | 1.3a | 3.5a | 2.0a |
| LSD | | 14 | 2.6 | 2.4 | 4.5 |

* The same letter indicates no statistically significant differences between locations with Fisher's LSD test ($p < 0.05$).

### 3.4. Runoff and Soil Erosion

Decreasing tree canopy cover in agroforestry systems significantly increased the surface runoff/rainfall ratio or the surface runoff/throughfall ratio (Table 4). In these results, the relationship between surface runoff with rainfall or throughfall was in line, and the ratio of surface runoff/rainfall was further used. The ratio of surface runoff/rainfall is also known as the runoff coefficient.

**Table 4.** Rainfall, runoff, ratio runoff/rainfall, and soil erosion in the runoff plots in each land cover type.

| Code | Land Cover | Rainfall (mm) | Runoff (mm) * | Runoff/ Rainfall Ratio * | Runoff/ Throughfall Ratio * | Soil Erosion (Mg ha$^{-1}$) * |
|---|---|---|---|---|---|---|
| *Upstream Rejoso Watershed* | | | | | | |
| UT1 | Old production forest | 555 | 14.3a | 0.03a | 0.04a | 5.86a |
| UT2 | Young production forest | 492 | 13.2a | 0.03a | 0.03a | 1.47a |
| UT3 | Agroforestry | 476 | 203.3b | 0.43b | 0.56c | 120.98b |
| UT4 | Arable land | 556 | 225.7b | 0.41b | 0.43b | 163.22b |
| LSD | | | 46.3 | 0.09 | 0.11 | 87 |
| *Midstream Rejoso Watershed* | | | | | | |
| MT1 | Old Production Forest | 616 | 80.2a | 0.13a | 0.16a | 3.07a |
| MT2 | Agroforestry | 841 | 316.3c | 0.38b | 0.48b | 2.88a |
| MT3 | Agroforestry | 616 | 228.8b | 0.37b | 0.46b | 6.63ab |
| MT4 | Agroforestry | 541 | 344.9c | 0.64c | 0.66c | 10.33b |
| LSD | | | 86.6 | 0.12 | 0.12 | 4.22 |

* The same letter indicates no statistically significant differences between locations with Fisher's LSD test ($p < 0.05$).

Infiltration rates in the andisols of the upper watershed were all above 45 mm hour$^{-1}$ (Figure 6a). In the midstream area, forest plots had a high infiltration rate, but in the agroforestry systems infiltration

rates were low and the apparent declined with decreasing tree cover was not statistically significant (Figure 6b).

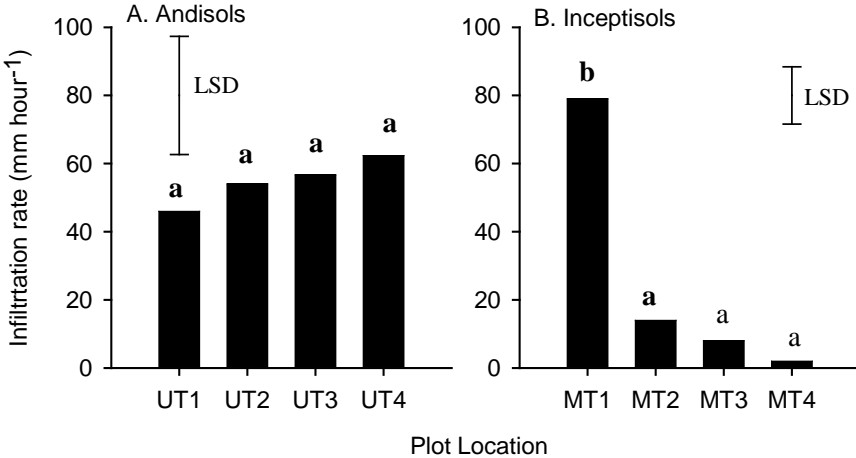

**Figure 6.** Soil infiltration rate measured using a double-ring infiltrometer (n = 6).

In the upstream area, with decreasing tree canopy cover, the surface runoff/rainfall ratio increased 16-fold compared to production forest (Figure 7a). In the midstream area, agroforestry systems with a tree canopy cover > 80% were still able to support low surface runoff (Figure 7b). With a tree canopy cover of < 80%, surface runoff increased rapidly on days with moderate rainfall (20–50 mm day$^{-1}$) (Figure 7b).

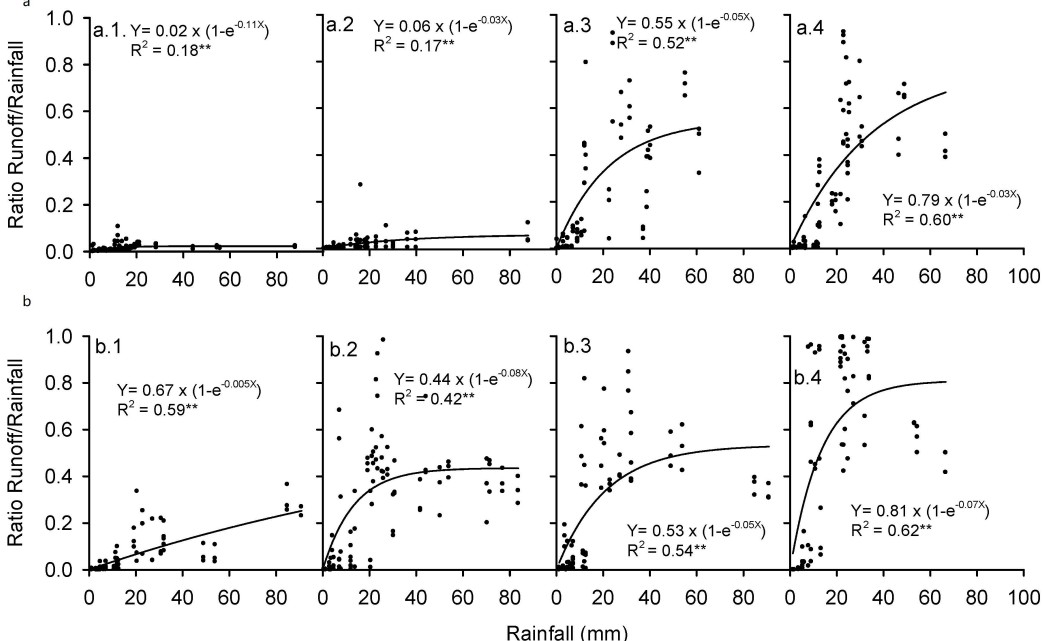

**Figure 7.** The relationship between surface runoff/rainfall ratio and the amount of rainfall in production forest and agroforestry systems in (**a**) the upstream Rejoso Watershed, under (a.1) 55% canopy cover of pine-based old production forest, (a.2) 40% canopy cover of pine-based young production forest, (a.3) 5% canopy cover of *Casuarina*-based agroforestry with cabbage crop, (a.4) 0% tree canopy cover of arable land (maize crop); (**b**) the midstream Rejoso Watershed under (b.1) 87 % canopy cover of pine/mahogany-based old production forest, (b.2) 75% canopy cover of coffee-based agroforestry, (b.3) 52% canopy cover of clove-based agroforestry, (b.4) 26% canopy cover of mixed tree and crop-based agroforestry.

In production forests with a closed tree canopy cover, soil erosion rates were low (Table 4 and Figure 8a.1,a.2,b.1). These production forests still had a protective understory vegetation that contributed to litter necromass and surface roughness (Table 3), controlling splash erosion. Upstream, with a reduction in tree cover, canopy soil erosion increased dramatically from 20 to 110 times the rates measured in forested plots (Table 4). Erosion rates in all plots increased with the amount of rainfall (Figure 8a.3,a.4). Midstream agroforestry systems had erosion rates ranging from 2.8 to 10.3 Mg ha$^{-1}$ in the measurement period (Table 4). As annual rainfall is approximately three times what was recorded in the measurement period, with similar rainfall intensities, these erosion rates are to be multiplied by a factor of three, leading to 9–31 Mg ha$^{-1}$ year$^{-1}$.

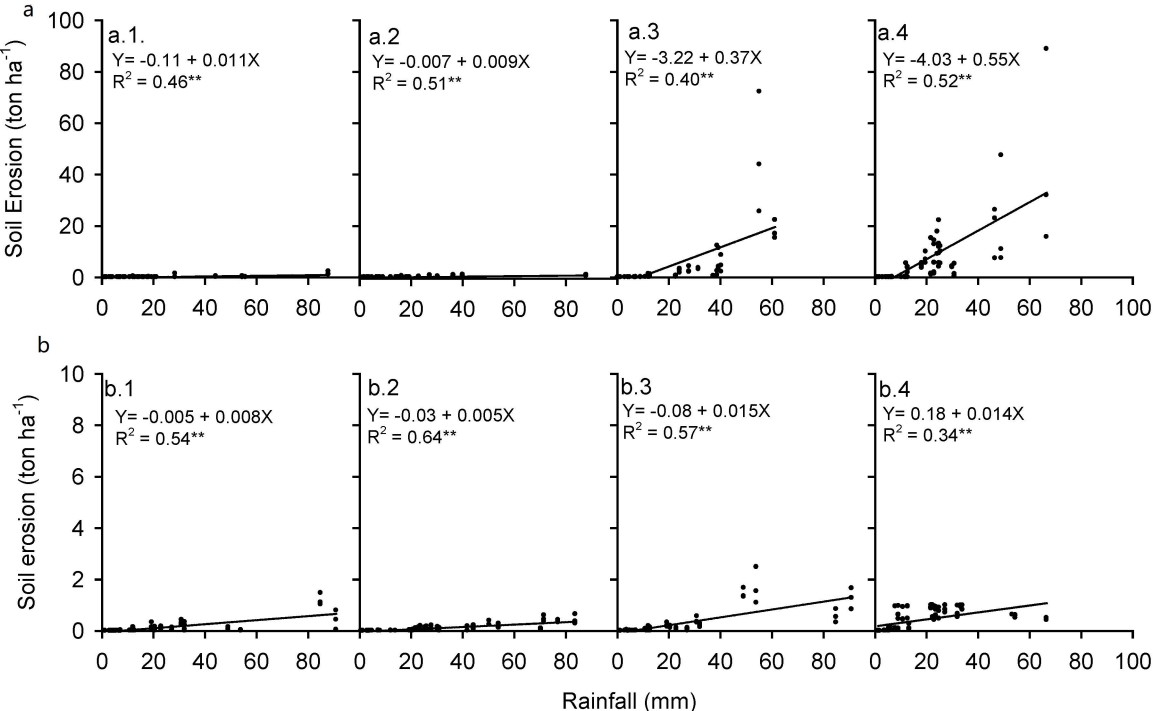

**Figure 8.** Soil erosion in relation to daily rainfall rates in production forest and agroforestry in (**a**) the upstream Rejoso Watershed, under (a.1) 55% canopy cover of Pine-based old production forest, (a.2) 40% canopy cover of pine-based young production forest, (a.3) 5% canopy cover of *Casuarina*-based agroforestry with cabbage crop, (a.4) 0% tree canopy cover of arable land (maize crop); (**b**) the midstream Rejoso Watershed under (b.1) 87% canopy cover of pine/mahogany-based old production forest, (b.2) 75% canopy cover of coffee-based agroforestry, (b.3) 52% canopy cover of clove-based agroforestry, (b.4) 26% canopy cover of mixed tree and crop-based agroforestry.

### 3.5. Thresholds for Infiltration-Friendly Land Use

Increasing tree canopy cover, while maintaining understory vegetation and litter necromass, is a strong indicator of watershed health and the main driver of low surface runoff (or high soil infiltration) and low soil erosion in production and agroforestry forest systems in the Rejoso Watershed (Figures 8a.1, 9a.1, 10b.1 and 11b.1).

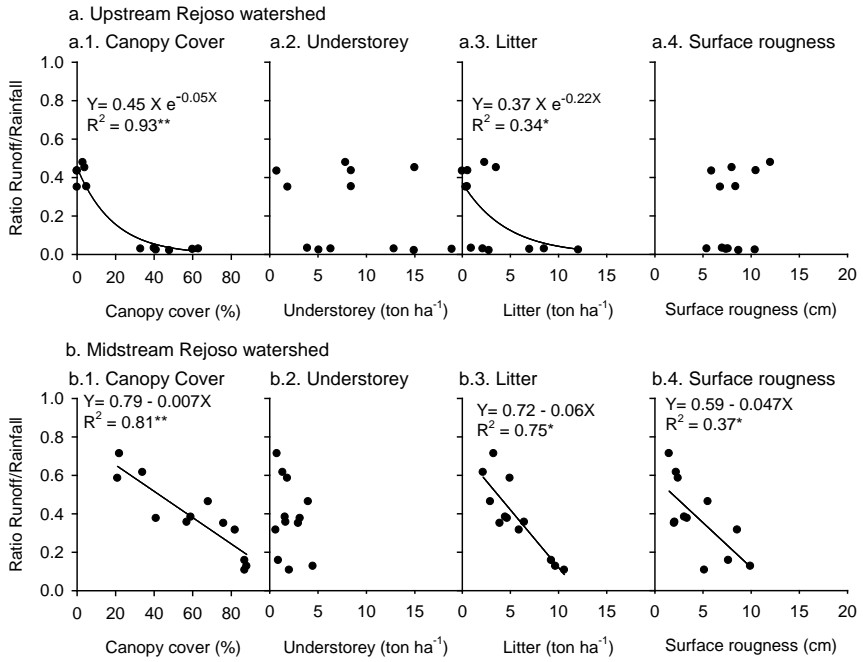

**Figure 9.** The runoff/rainfall ratio as a function of tree canopy cover, understory vegetation, litter necromass, and land surface roughness in the (**a**): upstream, (**b**): midstream.

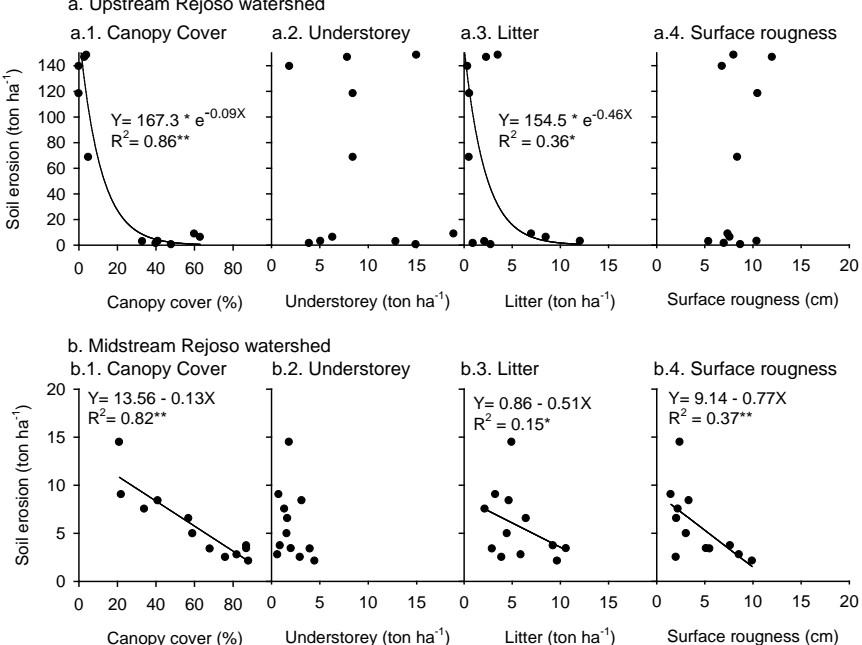

**Figure 10.** Soil erosion in relation to tree canopy cover, understory vegetation, litter necromass, and land surface roughness in the (**a**): upstream, (**b**): midstream.

Understory vegetation theoretically can reduce splash impacts on the soil and supports infiltration, as does the litter necromass present. However, the result of this study indicated that understory had no statistically significant relationships with runoff coefficient and soil erosion (Figures 9a.3,b.3 and 10a.3,b.3). Land surface roughness, in contrast to litter necromass, had no consistent relationship with runoff or erosion (Figures 9a.4 and 10a.4).

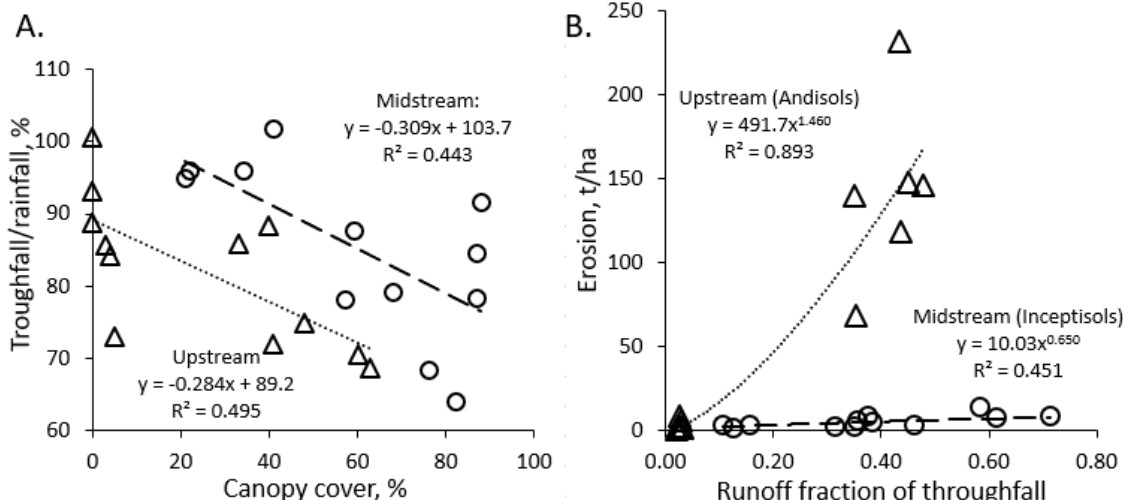

**Figure 11.** Comparison of canopy effects on throughfall in the two zones (**A**) and the relationship between erosion and surface runoff for the two soil types (**B**).

Summarizing (Figure 11), we found a similar slope (3% more canopy water retention per 10% canopy cover) but a 10% higher canopy retention overall in the upstream area (with lower rainfall intensity and more small events) and a strong difference between the two soil types in erosion per unit surface runoff, offsetting the higher infiltration in Andisols.

## 4. Discussion

Our study only covered two months' worth of data, rather than the recommended 3–5 years for such studies. Measurements included one-fourth of the mean annual rainfall, and the validity of the result may be primarily limited by the assumption that it represented a fair sample of the rainfall intensities that can be expected in the landscape. With a disproportional fraction of erosion normally associated with extreme events (compare the curvature of responses in Figure 11B), scaling up our comparison among land cover types to a multi-year basis may underestimate the relevance of controlling overland flows.

The first research question tested the hypothesis that, along the forest to open field agriculture continuum, there is a significant decrease in soil hydrological functions. The results of the present study confirmed that the conversion of high-density forest to land uses with a lower tree canopy significantly decreased soil infiltration rates (Table 2). The results of this study align with previous studies that showed that decreases in ground cover resulted in decreases in soil infiltration rates [51]. Forests and coffee agroforestry have been shown to reduce surface runoff and erosion compared with coffee monoculture [52]. Soil infiltration into Andisols both under deciduous and pine forest was higher than that in cropland in a study on the Canary Islands [53]. A study in China [54] found that the soil infiltration rate of forest was greater than that of agroforestry. A meta-analysis [55] concluded that converting any land use type with permanent vegetation cover (grassland, shrub, or forest) to seasonal cropland leads to a decline in the soil infiltration rate, harming soil and water conservation, while agroforestry improved the soil infiltration capacity compared to cropland and plantations.

The degradation of the soil hydrological functions of forest could be attributed to the decrease in soils' macroporosity, organic matter content, and increased soil bulk density (Table 2), which had relevance to the decreasing infiltration rates (Figure 8). Among various land use patterns, plant root activities are important factors affecting soil infiltration [56]. The reason why cropland has a lower infiltration rate than the land use types with a high density of trees compared with those with a low density of trees in forest may be verified by the fact that soils beneath the canopies of woody plants had a more extensive distribution of plant roots and a greater number of macropores, which are biologically produced pores [57,58], which created a positive feedback on infiltration [59,60]. The soil

macroporosity, needed for effective infiltration, is the result of a continuous process of compaction and the filling in of macropores with fine soil particles, and the creation of biogenic channels (formed by old tree roots, earthworms, and other soil engineers) or abiotic processes (cracks). As no heavy machinery is used in any of these land use systems, compaction is restricted to human feet and motorbikes on specific tracks. The formation of old tree root channels can cause long time lags between land cover change and soil macroporosity [61,62], obscuring relations between current tree cover and soil hydrologic functions. "Fallows" were found to be intermediate between forests and grasslands in terms of infiltration in Madagascar [63]. Recovery of infiltration after the reforestation of grasslands in the Philippines was found to be a matter of decades rather than years [64]. In studies elsewhere in Indonesia, forest soils had more macropores and higher surface infiltration rates than monoculture coffee plantations [52]. Land use changes, especially from forest to cropland, have caused remarkable changes in soil properties, including the loss of organic matter and increases in bulk density [65], which lead to decreased infiltration rates [66]. Some researchers suggested a positive relationship between soil organic matter and infiltration rate [67,68].

Our study results can be compared to earlier studies in the volcanic uplands of West and East Java. Sediment delivery to streams increased after a clear-felling and replant operation in the Citarum Basin (W. Java), which involved the delayed flushing of material trapped during forest clearance and the incipient gullying of trails created by farmers involved in the replanting program, rather than by in-field erosion [69]. A study [70] of the Kali Konto catchment in East Java similarly concluded that "despite their relatively small areal extent (5% in the study area), rural roads, trails and settlements are significant producers of runoff and sediment at the catchment scale and should be included in watershed management programs designed to reduce catchment sediment yields and reservoir siltation." Soil conservation practices that transform slopes to relatively flat terrace beds and steep terrace risers are, in the absence of vegetation, still subject to erosion, with splash impacts on the terrace risers as a major cause [71,72].

The second research question came out with the hypothesis that the dominant factors that determine "infiltration friendliness" at the plot scale are tree canopy cover, understory vegetation, litter necromass, and land surface roughness. Our research shows that a number of land cover types had infiltration rates below the required rates at peak rainfall events. Among the four factors tested, tree cover and litter layer necromass could be used to define zone-specific thresholds for infiltration-friendly land use, but understory vegetation and surface roughness could not. Although slopes in the upper watershed are much steeper than in the midstream, the coarser texture and likely higher aggregate stability means that thresholds for canopy cover and litter necromass can be lower. A first "line of defense" of forests is the canopy retention of rainfall, prolonging the time for infiltration, as canopy dripping lasts beyond the rainfall event. Canopy retention of rainfall tends to be relatively high for small (but potentially frequent) rainfall events, and low for high rainfall intensities. Our throughfall results for the two zones corresponded with differences in observed intensity. A five-year study in the Amazon forests of Colombia [73] showed that throughfall ranged from 82 to 87% of gross rainfall in the forests studied (with a canopy cover of 83–91%) and varied with event-level gross rainfall, but also with forest structure, while stemflow contributed, on average, only 1.1% of gross rainfall in all forests. Throughfall is more spatially heterogeneous than rainfall, creating a challenge for its measurement. Roving, rather than fixed, location throughfall gauges led to narrower confidence intervals of throughfall fractions in longer-term studies [74] in lower montane rainforest in Puerto Rico, where throughfall was 75% and stemflow 4.1% of rainfall, with palms responsible for about 3% and other trees 1.1%. Spatial heterogeneity in throughfall can be expected to lead to uneven patterns of deep percolation and groundwater recharge in "patchy" forests [75]. Canopy interception can lead to direct evaporation, throughfall, or stemflow [76]. The ratio between throughfall and stemflow depends on the architecture of leaves (e.g., erect leaves favoring stemflow, pendulous leaves favoring throughfall) and stems. Storage along the stem pathway depends on bark properties [77]. Stemflow accounted for less than 3% of gross rainfall for tropical hardwoods in a study in Panama, while it was high for tall

grasses [78]. High stemflow fractions have also been reported for bamboo, bananas, shaded coffee and cocoa, and understory shrubs [79–82]. Canopy interception and direct evaporation tend to be high in coastal areas with frequent light rainfall events, but low where tropical rainstorms are predominant and the canopy storage is rapidly saturated [83,84]. By creating throughfall drops that are larger than those of open-field rainfall, tree canopies may increase sub-canopy erosivity [13,84].

Many authors have emphasized that the key to hydrologic functions is in the soil rather than the aboveground parts of the forest [12]. Still, we found strong and direct relations with canopy cover. Positive effects of canopy cover on infiltration were related to raindrop interception in earlier studies [75]. Interception will (a) reduce the destructive power of rainwater splash on the ground surface (as long as the erosive canopy drips described earlier are avoided), (b) allow more time for infiltration as water reaches the surface more slowly, (c) keep a thin water film on the leaves that will (d) cool the surrounding air when it subsequently evaporates. It reduces the amount of water reaching the soil surface, but by increasing air humidity it also decreases transpiration demand when stomata are open. Coffee gardens close to forest had high macroporosity and infiltration rates relative to more compacted pasture and sugarcane land on volcanic slopes in Costa Rica [85]. Dye infiltration patterns in a comparison of natural forest and rubber plantations in Yunnan (China) showed [86] that the fine roots of understory vegetation promoted subsurface flow and reduced water erosion. The effects of trees on infiltration have been described as a "double-funneling" [87] with stemflow (dependent on the insertion angle of branches on the main stem), bringing water to the soil surface connection point for root-induced preferential flow [88,89].

A comparison of infiltration rates (median $K_s$ values 16–98 mm h$^{-1}$) in broadleaf, pine-dominated, and mixed community-managed forest in Nepal [90] found the less intensively used pine-dominated site to be more conducive to vertical percolation than the other two forest types. These results were remarkable in relation to the negative local perceptions of the role of pine plantations on declining water resources.

Understory vegetation can theoretically reduce splash impacts on the soil and supports infiltration, as does the litter necromass present. However, the result of this study indicated that the understory shows no significant relationships with the runoff coefficient and soil erosion. This is possibly because surface runoff and erosion are largely controlled by land cover. The growth and development of the understory is determined by canopy cover. Likewise, the tree plantations in each plot are also diverse, so this also affects the diversity of the understory vegetation underneath. The result of this study indicates that the litter layer in the old production forest both upstream and midstream is significantly thicker than that other land uses (Table 3) and there is a significant correlation with the runoff coefficient and soil erosion (Figures 7 and 8, respectively). Litter is the parts of the body of the plant (in the form of leaves, branches, twigs, flowers, and fruit) that die (deciduous or pruned) and lie on the surface of the soil either intact or partially weathered. The role of litter in maintaining infiltration and soil erosion is through: (a) M=maintaining soil looseness by protecting the soil surface from rainwater, so that aggregates and soil macropores are maintained, (b) providing food sources for soil organisms, especially "soil engineers" (e.g., earthworms), so that the organism can live and develop in the soil, thus, the number of macro pores is maintained through the activity of these organisms, and (c) maintaining water quality in the river through the filtering of soil particles carried by surface runoff before entering the river. In a study in North China [91], the presence of the litter of *Quercus variabilis*, representing broadleaf litter, and *Pinus tabulaeformis*, representing needle leaf litter, reduced surface runoff rates by 29.5% and 31.3%, respectively. The overall effect of fast plus slow decomposing surface litter means the protection of the soil surface from splash erosion, surface roughness that reduces sediment entrainment, an energy source for soil biota, and a conducive microclimate [92,93].

Infiltration fractions depend on the scale of measurement and on variations in slope steepness, as overland flow can re-infiltrate on less steep foot-slopes in the case of the upper plots [94] or water infiltrates can re-emerge as surface flow depending on subsoil conductivity [95,96]. Such effects will need to be included if catchment level hydrology is to be predicted from plot-level measurements.

The land surface roughness also contributes to a high infiltration rate, reducing soil erosion. In the upstream, there is no significant different between land uses, but in the midstream, land surface roughness in agroforestry systems with tightly different canopies is significantly higher than rare canopies (Table 3). Without a high canopy cover (Table 3), this roughness was not able to control surface runoff and erosion in the upstream area. This is due to a steep slope in this plot. Both the production forest and agroforestry systems with high canopies maintained a relatively high land surface roughness compared with rare canopies in the midstream area. In the midstream, the land surface roughness was significantly correlated with the runoff coefficient and soil erosion. The role of surface roughness as a sediment filter may depend on frequent regeneration to counter homogenization [97]. Surface roughness in the landscape includes a cavity, the meandering of streams due to the presence of litter, necromass, tree trunks, and rocks, which provide opportunities for water flow to stop for longer periods and experience infiltration. This condition also functions as a sediment filter. This function needs to be managed through land management, so that surface roughness is maintained on the ground.

Shifts in local rainfall patterns between sub-watersheds make it difficult to disentangle the relative importance of land use and climate change through statistical pattern analysis without knowledge of the underlying processes [98,99]. The holy grail of scientific hydrology, connecting overall aggregated flow patterns to local extreme events and possible hysteresis, is still worth searching for even if a general solution might ultimately prove impossible to find [100]. For the deep seepage component of the hillslope and catchment water balance, we can expect that extreme events are less important than gradual changes that influence average flows, but empirical analysis of the uncertainties involved is still a challenge [101].

The third research question is, as an analysis, the answers to the previous two research questions with the hypothesis that it is not always that the upstream watershed area is more sensitive to hydrological disturbance due to changes in land use than the midstream, but the factor of soil properties also determines considerations in watershed hydrological management. From a land use policy perspective, our results suggest that maintaining high (~80%) canopy cover in the mid-slope farmer-controlled landscape under bench terracing, which does not match the slope criteria for designation as watershed protection forest, is important. In Indonesia, protection forest areas have the primary functions of the protection of life support systems to regulate water management, prevent flooding, control soil erosion, and maintain soil fertility [102].

Erosion rates of 9–31 Mg ha$^{-1}$ year$^{-1}$, as estimated here, are a challenge, especially if the 400-year time frame of using up all soil, as used in Equation 7, is replaced by a tolerance equal to the rate of soil formation. A study in a high rainfall area with Inceptisols in Central Java [103] estimated that the rate of chemical weathering was around 0.85 Mg ha$^{-1}$ yr$^{-1}$ and used that as estimate of erosion rates that can be sustained indefinitely without affecting soil depth. Volcanic ash inputs add soil on top of the profile but may also be disproportionately included in what gets removed from the plots. Our measurements in Rejoso suggested that critical thresholds of the degree of canopy cover that is hydrologically desirable depend on soil and climatic conditions, which may vary over a relatively short distance. When the focus is on erosion and net sediment transport, the scale of consideration strongly influenced conclusions in the volcanic Way Besai Watershed in Sumatra as well [104]. With the higher rainfall intensities in midstream Rejoso and more erodible soils upstream, the risks for degradation from a downstream perspective are differentiated by zone. Combining our plot-level results with efforts of hydrologic modeling for the Rejoso catchment as a whole [105,106] can guide further advice to a local watershed forum on the measures and incentives needed to restore and protect the watershed as a whole.

The Indonesian legal requirement of 30% forest cover across all its local government entities [31] is a coarse translation of the hydrologic relations at risk. It clearly matters what the land cover in the "non-forest" parts of the landscape is and how vegetation interacts with soils and geomorphology in shaping rivers and groundwater flows [107,108]. Our findings for the Rejoso Watershed show that, within the agroforestry spectrum, hydrologic the thresholds of infiltration friendliness exist between the systems that are mostly "agro" and those that are mostly "forest", but higher tree cover systems are desirable.

## 5. Conclusions

Our results demonstrated that vegetation-based thresholds for adequate infiltration, given the existing rainfall intensities, differed between the middle and upper Rejoso Watershed. Despite steep slopes and low tree cover, the upper watershed, with its course soil texture (pseudo-sand/silt), low bulk density due to a high content of amorphic minerals, strong micro-aggregation and individual minerals, sponge-pores typical of Andosols, and land management practices that combine vegetable crops with a tree canopy cover of around 55%, can maintain infiltration and keep erosion at acceptable levels. In the midstream part of the catchment, despite gentle slopes under bench terracing, infiltration-friendly land use on the fine-textured Inceptisols required a canopy cover of 80%. Beyond tree canopy cover, litter layer necromass was found to be a good and easily observed indicator of infiltration rates, while understory vegetation and surface roughness may support infiltration, but are not sufficiently strong indicators.

**Author Contributions:** D.S., W.W., K.H., and M.v.N. designed the study. N.M. collected data in the midstream, A.L.R. collected data in the upstream, R.M.I. coordinated the data collection in the field, and were academically supervised by D.S., W.W., and K.H. D.S., K.H. and M.v.N. shaped the manuscript, which was approved by all co-authors.

**Funding:** Fieldwork for this research was funded by the Danone Ecosystem Fund via the World Agroforestry Centre, ICRAF, Indonesia office. This research was **also** partially funded by the Indonesian Ministry of Research, Technology and Higher Education under the WCU Program managed by **the** Institute of Research and Community Services, Universitas Brawijaya and Institut Teknologi Bandung.

**Acknowledgments:** Authors thank to the community of the Rejoso Watershed and the "*Rejoso Kita*" Forum. They also thank the Department of Soil Science, Faculty of Agriculture, University of Brawijaya and the Research Group of Tropical Agroforestry for their support of the research. The authors would like to thank the Social Investment Indonesia (SII) organization for connecting the local stakeholders during fieldwork. The manuscript benefitted from the comments of Sampurno Bruijnzeel and an anonymous reviewer.

**Conflicts of Interest:** The authors declare no conflict of interest.

## Appendix A

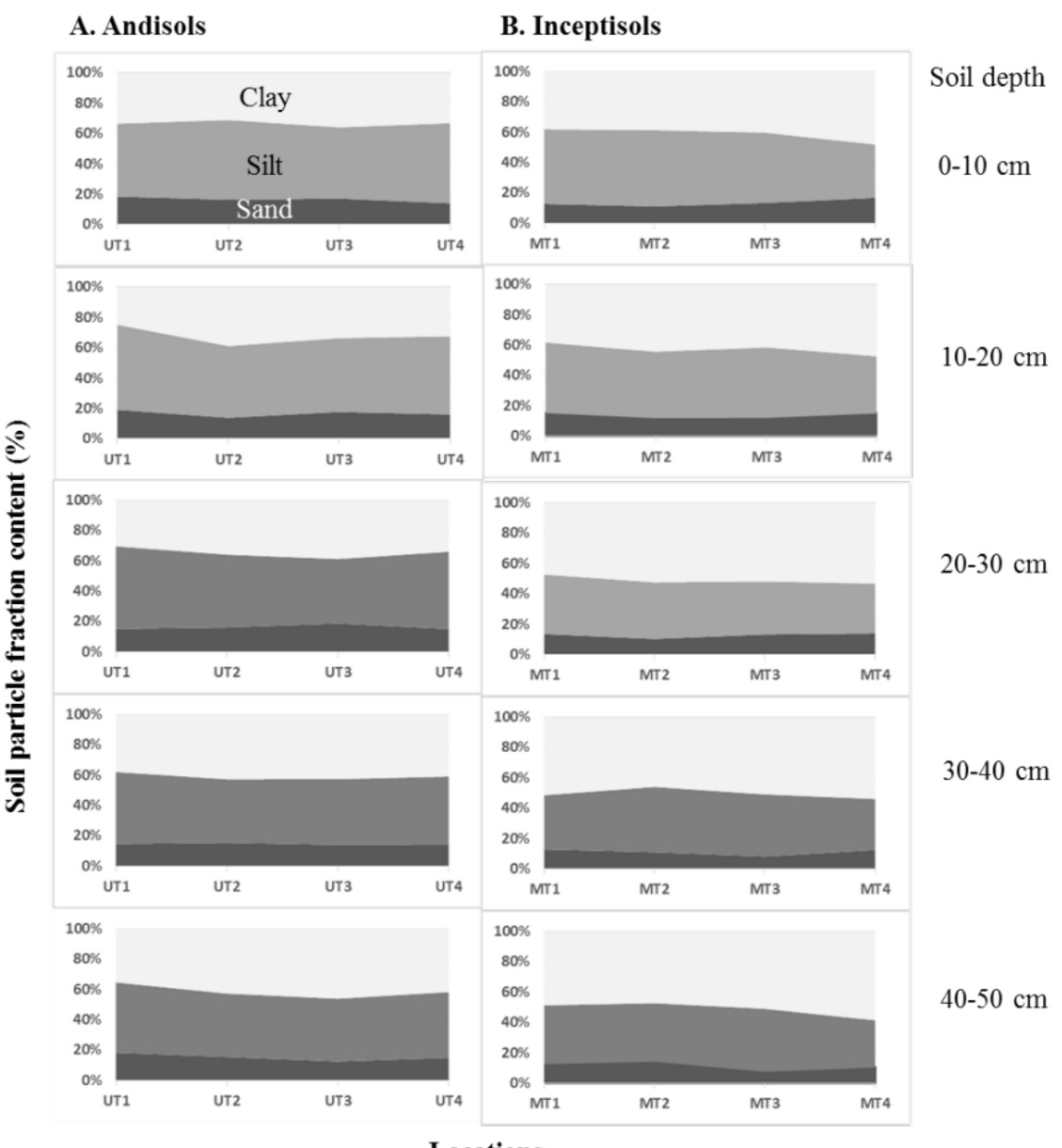

**Figure A1.** Soil texture in five different layers in runoff plot measurements.

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
