# Peer review of "Infiltration-Friendly Agroforestry Land Uses on Volcanic Slopes in the Rejoso Watershed, East Java, Indonesia"

_land, doi:10.3390/land9080240_

Round 1

Reviewer 1 Report

My worry is that data was collected during 2 months, which I think does not represent rainy season length in the area of study; also, the should provide a control (bare land I think), and compare the treatments with the control

Reviewer 2 Report

The article is well written and makes an important contribution to the field. The research depth is sufficient, and the approach is well grounded in the international literature. There are few five tuning suggestions which can improve the presentation.

1. The abstract presents in too much detail the results, including unnecessary figures (values), and does not say anything about their significance, which is properly presented in the "Discussion" section. The unnecessary details should be eliminated and replaced with 1-2 sentences describing the significance of the results.
2. Fig. 1 shows the location of the case study in a very local context, and cannot be understood by the broad international research audience of "Land". A new separate figure is required, showing the position of the case study in a broader, worldwide context, e.g. the position of the volcano (showed as a dot) in Indonesia, including some neighboring countries.

Reviewer 3 Report

Abstract: What are the implications of findings? need to be briefly added.

Introduction was sound and coherent with clear research questions and objectives.

Methods: Figure 1 needs to be provided with a higher quality.

Results are properly presented.

Discussion was interesting. Adding the limitations of methodology/data and how they influence on the results would be help readers better understand this research.

Reviewer 4 Report

I have read and reviewed the manuscript entitled 'Infiltration-Friendly Agroforestry Land Uses on Volcanic Slopes in the Rejoso Watershed, East Java, Indonesia'. Overall I think it's a fine paper with important results which identify the capacity of various agricultural land covers to increase infiltration and the provision of water services in volcanic landscapes. I have a few othercomments below for the authors to consider to help improve the manuscript.

The paper was well-written, if a little wordy.  I recommend that the authors attempt to distill the amount of text by at least 20% overall.

In the last paragraph of the introduction the authors should expand on the objectives of the study including a brief description of the methods used to address the questions.

 the Colours in Figure 1 are rather awful on the eye. I understand that this might be some sort of false colour satellite image but I think the authors can do a better job such as overlaying a land-use map over the topographical Hill shade and provide a legend. I would also like to see the elevation mapped in this figure.

Figure 2 is too bold and could be improved. In fact all of the graphs have some sort of weirdness. Figure 5 has abnormally thick x-axis and tick marks and bold x-axis labels, figure 6  has axes that are too thick. This draws the attention to the graph elements not the actual data.  Figure 8 has a strange error bar above the actual data and  labels which are not explained in the caption. Make all graphs have a consistent look and feel.

Figure 3 needs to be completely redesigned as it is almost impossible to read or make sense of.

The manuscript needs a proof read as there are many typos such as double brackets misplaced spaces.

In figure 10 why isn't our value used and not R squared? Specially given R squared is used in figure 13. I suggest using R squared throughout.

 The number of figures and tables in this manner script is excessive and some should be moved to the supporting information.

 113 references is also somewhat excessive and could be reduced.
